# Look at My Body: It Tells of Suffering—Understanding Psychiatric Pathology in Patients Who Suffer from Headaches, Restrictive Eating Disorders, or Non-Suicidal Self-Injuries (NSSIs)

**DOI:** 10.3390/pediatric17010021

**Published:** 2025-02-08

**Authors:** Diletta Cristina Pratile, Marika Orlandi, Adriana Carpani, Martina Maria Mensi

**Affiliations:** 1Department of Brain and Behavioral Sciences, University of Pavia, 27100 Pavia, Italy; dilettacristin.pratile01@universitadipavia.it (D.C.P.); martina.mensi@mondino.it (M.M.M.); 2Child Neurology and Psychiatry Unit, IRCCS Mondino Foundation, Via Mondino 2, 27100 Pavia, Italy

**Keywords:** adolescence, restrictive eating disorders, NSSI, headaches, Rorschach, prediction, prevention, psychiatric profiles, multimethod assessment, functioning

## Abstract

**Background/Objectives**: Adolescence is a developmental stage characterized by profound physical and psychological transformations, often leading to vulnerabilities such as body dissatisfaction, identity challenges, and the use of maladaptive coping strategies. This often leads to body-related psychopathologies, including headaches, restrictive eating disorders, and non-suicidal self-injury (NSSI). The present study aimed to describe the typical functioning and features of these conditions and the differences between the three groups, and to identify the most effective assessment for predicting these conditions. **Methods**: Sixty adolescent patients (51 female; mean age = 15.34 ± 1.80) were divided into three groups: headaches, restrictive eating disorders, and NSSI, and assessed for differences in symptoms, cognitive performance, personality, functioning, and illness severity using semi-structured interviews, clinician-based scales, and performance-based tests like the Rorschach inkblot test, according to the Rorschach Performance Assessment System (R-PAS). **Results**: Individuals with headaches experienced more internalizing symptoms, had an average IQ, maintained some functioning areas, and had distorted patterns of self–other relationships with the tendency to project malevolent aspects onto others. Patients with restrictive eating disorders had high levels of depressive symptoms, above-average IQ scores, negative symptoms, moderate presence of obsessive–compulsive personality traits, disorganized thinking, and a tendency to interpret situations subjectively. Patients with NSSI showed the highest level of depressive symptoms and social anxiety symptoms, and a considerable presence of psychotic symptoms and perceptual distortions. Negative symptoms, borderline personality traits, and psychosis symptoms had the strongest predictivity. **Conclusions**: The study provides clinicians with relevant insights into the features of these conditions and highlights assessment strategies, tailored interventions, and enhanced outcomes for these vulnerable populations.

## 1. Introduction

Adolescence represents a critical developmental stage marked by significant physical and psychological changes, which can give rise to vulnerabilities such as body dissatisfaction, identity struggles, and the adoption of maladaptive coping mechanisms [1,2]. These vulnerabilities often manifest as psychological distress and emotional challenges expressed through the body. Such manifestations encompass a spectrum of psychopathological presentations varying in severity, intensity, and levels of awareness. Among these, restrictive eating disorders, non-suicidal self-injury (NSSI), and somatic symptoms such as headaches are frequently observed [3,4,5].

Headaches are a common somatic expression of psychological distress in adolescents. Recurrent episodes, often linked to elevated stress and emotional difficulties, significantly impact daily functioning and quality of life [6,7]. Eating disorders, particularly anorexia nervosa (AN), are prevalent and influenced by multiple interacting factors. Emerging during adolescence, these disorders frequently persist into adulthood, underscoring the urgency for early and targeted interventions [8,9,10,11]. NSSI, characterized by deliberate self-inflicted harm, often coexists with other psychiatric conditions and serves as a maladaptive coping mechanism for emotional pain expression [5,12,13].

In clinical practice, these three conditions frequently coexist, with patients often presenting overlapping symptoms and shared mechanisms of emotional dysregulation and somatic expression. Understanding these conditions’ unique characteristics and potential intersections is critical for improving assessment and tailoring intervention strategies. For instance, Davico et al. [14] found that adolescents with AN and NSSI displayed distinct psychopathological and cognitive profiles compared to those with AN alone, suggesting implications for treatment approaches. Similarly, migraine headaches have been associated with an increased risk of deliberate self-harm, although evidence linking migraines to suicide mortality remains inconclusive. Healthcare professionals are advised to monitor suicide risk in individuals with migraines [15]. Additionally, migraine symptoms have been linked to a higher likelihood of eating disorder symptoms in adolescents, particularly in females, highlighting the need for integrated screening and intervention strategies [16].

Given the transdiagnostic nature of symptoms, it is crucial to provide clinicians with reliable tools to accurately assess and understand adolescent psychopathology, thereby guiding appropriate care and ensuring timely interventions. However, no single assessment tool is sufficiently compelling. Thus, a multimethod assessment approach is preferred. These comprehensive assessments, incorporating self-reports, informant reports, clinical interviews, performance-based tests, and clinician-based scales, offer a holistic view of the adolescents’ resources and difficulties, favoring more targeted and effective interventions [17].

Within the multimethod assessment framework, the Rorschach test, administered using the Rorschach Performance Assessment System (R-PAS) [18], has proven to be a highly reliable and valid tool. It features up-to-date normative data that account for variations in age, gender, ethnicity, and education, as well as excellent inter-rater reliability [19,20]. This performance-based, hard-to-manipulate assessment evaluates psychological and psycho-legal functioning [21], an individual’s approach to reality and relationships, and implicit characteristics that the respondent may not consciously recognize [22].

The first aim of this study was to describe the typical functioning and features of three psychiatric conditions associated with physical manifestations of psychological distress: headaches, restrictive eating disorders, and NSSI.

The second aim was to highlight the differences between these conditions and identify overlapping features to provide clinicians with insights into shared and distinct mechanisms. Furthermore, this study explored which assessments are most effective in predicting the presence of these symptomatologic profiles, thereby enhancing diagnostic accuracy and treatment planning.

Considering previous research and theoretical considerations, we hypothesized that adolescents engaging in NSSI would exhibit the lowest levels of global functioning, given the strong association between self-injurious behaviors and broader psychosocial impairments. Moreover, we expected that the group affected by restrictive eating disorders would display higher IQ scores than the other groups, potentially reflecting the perfectionism and achievement-oriented traits often associated with this population. However, we also expected this group to show significant impairments in self- and other-representation, as interpersonal difficulties are commonly observed in adolescents with eating disorders. Finally, we proposed that adolescents suffering from headaches would demonstrate greater rigidity in emotional expression and regulation strategies. This expectation was based on the idea that somatic symptoms, such as headaches, may act as a manifestation of unexpressed or poorly regulated emotional states.

## 2. Materials and Methods

### 2.1. Study Design

This cross-sectional cohort study received approval from the Ethics Committee of Policlinico San Matteo in Pavia. It was conducted following the reporting of studies conducted using observational routinely collected health data (RECORD) statement (Appendix A). All patients and their caregivers provided written informed consent to participate in the study and were free to withdraw at any moment without explanation. The authors affirm that all procedures contributing to this research comply with the ethical standards of the relevant national and institutional committees on human experimentation, the Helsinki Declaration (1964), and its later amendments [23]. The data were pseudonymized and are available in the Zenodo repository upon request [24].

### 2.2. Study Population

From October 2019 to December 2023, we recruited 60 help-seeking adolescent patients (aged 12 to 18 y.o.) at the Child Neurology and Psychiatry Unit of the IRCCS Mondino Foundation in Pavia, Italy. These patients were either inpatients or on day hospital regimens. We divided the participants into three groups of 20 each. The first group consisted of patients diagnosed with headaches, the second group included patients diagnosed with restrictive eating disorders, and the third group comprised patients with a medical history of NSSI but without previous suicidal attempts. Inclusion criteria for the first group required a clinical diagnosis of primary headache according to the International Classification of Headache Disorders-3 [25]. The second group comprised patients presenting a diagnosis of any eating disorder with restrictive characteristics. We included restrictive subtypes of anorexia nervosa (AN), atypical anorexia nervosa (A-AN), avoidant/restrictive food intake disorder (ARFID), or other specified eating disorders with restrictive characteristics. All the diagnoses were made according to the Diagnostic and Statistical Manual of Mental Disorders (DSM-5) criteria [26]. Diagnoses were confirmed using the DSM-based semi-structured interview Kiddie Schedule for Affective Disorders and Schizophrenia, Present and Lifetime (K-SADS-PL-DSM-5) [27]. Finally, a positive medical history of NSSI, without a history of suicidal attempts, was needed to be assigned to the third group.

Exclusion criteria were: (I) intellectual disability (IQ ≤ 70) assessed using age-appropriate Wechsler intelligence scales (WISC-IV or WAIS-IV) [28,29], (II) insufficient understanding of the Italian language, (III) a history of head trauma or any other underlying medical or neurological condition, and (IV) substance use disorder. Additionally, participants in one group could not have comorbid diagnoses from the other two groups. Figure 1 shows the study population flowchart.

### 2.3. Procedures and Measures

Clinicians conducted a broad multimethod clinical assessment (see Figure 2), including the semi-structured interview Kiddie Schedule for Affective Disorders and Schizophrenia Present and Lifetime-DSM-5 (K-SADS-PL) [27,30], which evaluates the presence of diagnostic comorbidities. Moreover, the Structured Clinical Interview for the DSM-5-Personality Disorder (SCID-5-PD) [31] was conducted with the patients just after the initial self-report questionnaire. This semi-structured interview, administered as a screening tool to all the participants from 14 years on, provides categorical diagnoses/features (present or absent) of personality disorders based on DSM criteria. Additionally, patients were assessed using the Comprehensive Assessment of At-Risk Mental States (CAARMS) [32], a semi-structured interview that applies validated and reliable cut-offs to identify adolescents at risk for full-blown psychosis and attenuated psychosis and assesses the presence of negative symptoms. We also considered the CAARMS negative symptoms subscale score, evaluating the presence of negative symptoms if the patient reported a score equal to or above three in at least one of the subscale’s domains (alogia, avolition/apathy, anhedonia).

Clinicians also evaluated the overall disease severity with the clinician report Clinical Global Impression Severity (CGI-S) [33]. They assessed the patient’s general functioning using the Children’s Global Assessment Scale (CGAS) [34] and the Social and Occupational Functioning Assessment Scale (SOFAS) [35]. Furthermore, to exclude the presence of intellectual disability and measure IQ, we employed the maximum performance tests Wechsler Intelligence Scale for Children (WISC-IV) [28] or the Wechsler Adult Intelligence Scale (WAIS-IV) [29], depending on the patient’s age.

Finally, the personological structure was explored by administering and scoring the typical performance Rorschach inkblot test according to the Rorschach Performance Assessment System (R-PAS) [18]. R-PAS is a performance-based test involving a perceptual task. The clinician administers ten cards, and the scoring allows the detection of the person’s functioning and personality features that even the respondent may not consciously recognize. The R-PAS method is reliable and statistically valid, reflects international evidence, and has up-to-date normative data [19,20].

### 2.4. Statistical Analysis

We conducted analyses using SPSS version 30.0.0.0 [36]. Descriptive statistical analyses were performed for demographic and clinical characteristics of the total sample and separately for each group. Statistical comparisons between the three groups completed descriptive analyses. The significance level was set to α = 0.05. Given the small sample size, we used the nonparametric Kruskal–Wallis test, complemented by post hoc analyses (Dunn test), to compare the three groups. Bonferroni correction was applied to all post hoc analyses to reduce the probability of type I errors due to multiple testing. Moreover, we reported fixed omega squared (ω^2^), a robust effect size measure suitable for small samples [37]. Thresholds indicate very small (<0.01), small (between ≤0.01 and <0.06), medium (between ≤0.06 and <0.14), and large effect (≥0.14). Finally, to explore the predictivity properties of the tests, we converted the K-SADS and SCID-5 PD scores into dummy variables (0: absence of symptoms, 1: presence of symptoms) and ran a linear regression [38]. Because of the absence of previous research supporting our hypotheses, we used the ANOVA statistically significant variables to run the model.

## 3. Results

We enrolled 60 participants (51 female; mean age = 15.34 years; SD = 1.80), divided into three groups: 20 patients reporting headaches (14 female; mean age = 15.02 years; SD = 2.01), 20 with a diagnosis of any restrictive eating disorder (19 female; mean age = 15.31 years; SD = 1.74), 20 presenting NSSI (18 female; mean age = 15.69 years; SD = 1.66). They did not statistically differ in age, t(2) = 2.302 *p* = 0.499, or gender, t(2) = 2.871 *p* = 0.065.

Means, standard deviations, the comparison of K-SADS-PL, SCID-5-PD, Wechsler scales, CAARMS negative symptoms scale, CGI-S, CGAS, and SOFAS scores between the groups and post hoc analyses are shown in Appendix A. Table 1 shows statistically significant comparisons, effect sizes, and post hoc analyses corrected using Bonferroni.

Appendix A shows the means, standard deviations, and comparisons between all R-PAS indexes in the three groups, and post hoc analyses. The captions of all R-PAS variables are shown in Appendix A. Table 2 shows statistically significant comparisons between groups in R-PAS indices, effect sizes, and post hoc analyses corrected using Bonferroni.

Furthermore, the hierarchical linear regression analysis examines the predictive effects of negative symptoms, borderline personality traits, and psychosis symptom severity on the group (Table 3). In Model 1, negative symptoms accounted for 31.5% of the variance in the dependent variable. In Model 2, borderline personality traits increased the variance to 63.9%. Finally, in Model 3, the addition of psychosis symptom severity further improved the model, explaining 68.8% of the variance.

Table 4 presents the stepwise linear regression analysis results to identify the most significant predictors of being in one of the groups. The dependent variable is the group, and three models are developed to include progressively more predictors. The regression analysis highlights the hierarchical importance of the predictors, with negative symptoms consistently showing the strongest association across all models, followed by SCID-5 PD borderline personality traits and, to a lesser degree, K-SADS-PL psychosis symptoms. These findings underscore the relevance of negative symptoms and borderline features in understanding group differences while also indicating a potential, albeit smaller, role for psychotic symptoms.

## 4. Discussion

The study’s first aim was to describe the typical functioning and features of three psychiatric conditions associated with physical manifestations of psychological distress: headaches, restrictive eating disorders, and NSSIs.

Half of the adolescents with headaches presented depressive, panic, separation anxiety, and generalized anxiety symptoms, along with mild OCD symptoms, consistent with previous studies [39,40,41,42,43,44]. Notably, over half of the patients with migraines met the criteria for at least one anxiety disorder during their lifetime [45,46,47], while psychotic symptoms were absent in this group. Personality assessments revealed minimal borderline and obsessive–compulsive traits, in line with studies highlighting emotional rigidity and a tendency to repress anger in adolescents with headaches [48], as well as higher hypochondriasis and hysteria traits in adulthood [49]. Psychologically, the tendency to repress emotions like anger or frustration might lead to somatization, manifesting physical symptoms such as headaches. Socially, familial dynamics and perceived expectations could reinforce maladaptive coping mechanisms, contributing to difficulties in expressing emotional needs. Their cognitive functioning was average, as confirmed by other research [50]. Despite mild clinical severity, these adolescents often maintain functioning in several areas, though their quality of life is notably lower than that of healthy controls [45]. While their average IQ and mild clinical severity suggest preserved cognitive abilities, their diminished quality of life highlights the hidden burden of their condition.

The Rorschach test revealed no significant perceptual or cognitive distortions but highlighted difficulties in self–other relationships, including a tendency to project negative traits onto others. Similar patterns have been observed in prior studies using the Comprehensive System (CS) [51], which noted challenges in emotion regulation and reliance on impulsive, maladaptive coping strategies [52,53]. The projection of negative traits onto others and impaired interpersonal interactions observed in Rorschach assessments might reflect a learned pattern of emotional rigidity and mistrust shaped by chronic pain and social withdrawal. Interventions targeting emotional awareness, adaptive coping, and family support are essential to improving outcomes in this population.

Furthermore, the restrictive eating disorders group reported a very high presence of depressive symptoms, mild social anxiety, low levels of psychotic and OCD symptoms, and a substantial presence of restrictive eating disorder symptoms, as assessed using the K-SADS-PL, consistent with previous studies [54,55,56,57,58]. This group also demonstrated average or above-average IQ scores across most indices, aligning with earlier findings [59]. Moreover, they exhibited moderate obsessive–compulsive personality traits [54] and a considerable presence of negative symptoms. Despite moderate-to-high clinical severity, they strive to maintain functioning, with only a few impaired areas reflecting persistent perfectionism common in patients with restrictive eating disorders [60]. Previous studies suggested that individuals with restrictive eating disorders and subthreshold psychosis presented more impaired functioning [61].

The R-PAS revealed more responses than the normative sample, alongside increased perception and thinking problems. This aligns with findings that patients suffering from restrictive eating disorders exhibit high rates of attenuated psychotic symptoms [61], such as dissociation proneness [62]. Research using the CS [51] with patients affected by AN similarly found increased disorganized thinking [56]. Our findings also highlighted severe misinterpretations or subjective interpretations of reality and reality-check dysfunctions. The literature indicates that patients suffering from restrictive eating disorders often struggle to identify common responses, displaying an individualistic, self-centered approach to perceiving and interpreting their environment [56,63].

This characteristic profile in patients with restrictive eating disorders can be primarily understood through psychological and social dynamics. Psychologically, the drive for perfectionism and an intense need for self-control often stem from a fear of failure and feelings of inadequacy [60,64]. These traits foster emotional suppression and rigid thinking patterns, which impair flexibility and adaptability in interpersonal relationships and self-perception. Difficulties in reality testing may arise from an overreliance on subjective interpretations and internal standards, further distancing these individuals from external perspectives [65]. Socially, cultural, and familial pressures emphasizing achievement, success, and idealized body image reinforce these tendencies, creating an environment that prioritizes external validation over emotional authenticity. Additionally, family dynamics characterized by high expectations, overprotection, or limited emotional expression can exacerbate perfectionistic behaviors and hinder the development of adaptive coping strategies [66,67,68]. Together, these factors contribute to a functioning profile marked by high internal conflict, emotional dysregulation, and a distorted sense of self and reality.

Finally, in the group of patients presenting **NSSI**, depressive symptoms were reported by all patients in the K-SADS-PL, confirming the previous literature [69], with a high prevalence of social anxiety symptoms, significant psychotic symptoms, and a low occurrence of OCD symptoms. This aligns with studies linking psychotic symptoms to NSSI events and suicide attempts in adolescents [70], while self-harm ideation correlates with an increased risk for psychotic experiences [71]. The group of adolescents with NSSI showed lower performance in working memory tasks, consistent with previous findings [72]. A high prevalence of borderline personality characteristics was identified, in line with a review highlighting longitudinal associations between NSSI and borderline traits [73]. Moderate avoidant and mild obsessive–compulsive personality traits were also observed, as noted in other adolescent studies [74,75].

Moreover, there was a significant presence of negative symptoms, severe disease intensity, and low global functioning, with impairments interfering with patient functioning, as reported in the previous literature [76]. This profile of functioning in patients presenting NSSI can be understood as a complex interplay of emotional dysregulation, interpersonal difficulties, and maladaptive coping mechanisms. Depressive and social anxiety symptoms, along with psychotic experiences, reflect underlying emotional distress that often manifests as self-destructive behaviors. The presence of borderline personality traits suggests an unstable self-image and difficulty in managing intense emotions, which may contribute to the frequent use of self-harm as a means of emotional regulation. Avoidant and obsessive–compulsive personality features further highlight challenges in interpersonal relationships and rigid thinking patterns, which may intensify feelings of isolation and exacerbate distress. The low level of global functioning and negative symptoms may indicate a significant impairment in social and occupational domains, making it harder for these individuals to engage with supportive systems or seek healthier coping strategies.

As other colleagues found, the Rorschach test revealed higher perception and thinking issues, particularly regarding reality perception and thought disorganization [77]. Studies using the CS [51] showed that high scores in perceptual distortion, cognitive slippage, and aggressive acts predict being in the self-harm group [78]. The group of patients with NSSI also exhibited subjective interpretations of situations and psychopathology [78], difficulty seeing others’ perspectives, and a more individualistic approach to interpreting reality, leading to frequent misunderstandings and isolation [79]. Rorschach findings support the notion that these patients perceive the world and their relationships in a fragmented and distorted manner, leading to misunderstandings and reinforcing their sense of isolation. Ultimately, this maladaptive profile can perpetuate a cycle of self-harm and emotional dysfunction, further entrenching these patterns over time.

The second aim was to explore the differences between those three psychiatric conditions to inform clinicians on prevention, assessment, and patients’ outcomes.

Data showed that the headache group had significantly lower depressive disorder scores on the K-SADS-PL compared to the restrictive eating disorder and NSSI groups, confirming the prevalence of depressive symptoms in these disorders [57,69]. The differences in depressive symptoms between the patients with headaches and the other groups were also supported by a large effect size, underscoring the clinical relevance of these findings. The group of patients exhibiting NSSI reported significantly higher psychotic symptoms than the group of patients affected by headaches, consistent with the previous literature [70,71]. However, CAARMS positive symptom scores showed no significant differences between groups, possibly due to sub-threshold scores for attenuated psychosis [32]. The group of patients with restrictive eating disorders exhibited the highest presence of OCD symptoms, significantly higher than the NSSI one, likely linked to perfectionism and high-functioning traits in patients with restrictive eating disorders. Additionally, the group comprising patients affected by restrictive eating disorder showed the highest prevalence of AN-related symptoms in K-SADS-PL, significantly higher than both NSSI and headache groups, highlighting distinct symptomatology. In the SCID-5-PD, there was a significant difference between the NSSI and headache groups, with the NSSI group showing higher borderline personality traits [73]. The NSSI group also performed worse than the patients with headaches on the working memory tasks from the Wechsler scales. The observed differences between the group that includes patients with headaches and those presenting NSSI corresponded to a medium effect size, reflecting nuanced yet clinically relevant cognitive processing and emotional regulation difficulties in the group of patients with NSSI. Both restrictive eating disorders and NSSI groups showed more negative symptoms on CAARMS compared to the headache group, with worse illness severity ratings. The CGAS showed differences in functioning mainly between headache and NSSI groups, while SOFAS revealed significant differences between restrictive eating disorder and NSSI groups. The R-PAS assessment revealed that the group including patients with headaches had significantly lower PHR/GPHR index scores than the groups of patients who exhibited restrictive eating disorders and NSSI. The PHR/GPHR index showed a medium effect size, reinforcing that patients in the headache group displayed less problematic self-representation than the restrictive eating disorder and NSSI groups [80]. These distortions were more prominent in the restrictive eating disorder and NSSI groups. The group of adolescents with restrictive eating disorders had significantly lower scores on W% and Dd%, supported by large effect sizes, indicating a tendency to focus on details rather than global features, aligning with previous studies on migraine patients. The group including adolescents with restrictive eating disorders also scored lower on the CBlend index than the NSSI group, suggesting emotional avoidance, while also showing higher scores on the C′ index and NPH/SumH index, indicating emotional reactivity and unrealistic interpersonal attributions. Conversely, the NSSI group scored higher on the PER index, indicating relationship defensiveness. A recent study suggested that narcissistic traits might protect against NSSI behaviors [81], while aggressiveness and perfectionism may mediate suicidal risk in self-harm adolescents [82].

Finally, the linear regression analysis highlighted the critical roles of negative symptoms, borderline personality traits, and psychosis symptom presence in predicting the group. The effect sizes calculated for group differences further underscore the transdiagnostic importance of these clinical features. For instance, the large effect sizes for variables like negative symptoms highlight their predictive relevance across groups, emphasizing the need for integrated interventions targeting these shared mechanisms. According to the model, negative symptoms consistently emerged as the strongest predictor. Borderline traits and psychotic symptoms, though less prominent predictors, also played crucial roles, particularly in distinguishing patients with NSSI. The transdiagnostic relevance of negative symptoms, borderline traits, and psychotic features as predictors underscores shared underlying mechanisms across these conditions. Addressing these issues through integrated treatments targeting emotion regulation, interpersonal functioning, and reality testing could reduce the hidden burden of these disorders.

This study highlights the distinct yet interconnected clinical profiles of adolescents with headaches, restrictive eating disorders, and NSSI, emphasizing the importance of a personalized, transdiagnostic approach to adolescent mental healthcare. Emotional rigidity and somatization in headache patients reflect maladaptive coping reinforced by family dynamics and social expectations, suggesting the need for interventions that foster emotional awareness and adaptive expression. Patients suffering from restrictive eating disorders, marked by perfectionism, reality distortions, and heightened emotional reactivity often rooted in cultural and familial pressures, benefit from therapies targeting cognitive flexibility and emotional authenticity. Adolescent patients with NSSI with severe emotional dysregulation, interpersonal difficulties, and psychotic symptoms require comprehensive treatments to address their multifaceted struggles. The study also underscores the critical role of family dynamics, advocating systemic interventions to improve communication and reduce parental overcontrol. Additionally, tools like the R-PAS and cognitive assessments reveal subtle impairments, offering valuable insights for tailored care. These findings call for early identification, prevention strategies, and holistic care models to support adolescents’ psychological, social, and familial needs, ultimately improving developmental outcomes.

This study has some limitations that should be considered when interpreting the findings. First, we excluded patients with more than one type of self-directed physical harm to reduce potential biases. This approach ensured that each clinical group represented distinct psychological profiles. Still, it also resulted in a relatively small sample size, which may limit the statistical power of our analyses and the ability to detect subtler effects. Future studies with larger samples would allow for a more robust exploration of these phenomena and improve the reliability of the findings. Second, the sample composition was unbalanced due to the overwhelming presence of female patients. While this reflects the higher prevalence of these conditions in adolescent females [83], the limited number of male participants restricts the generalizability of the results to the broader adolescent population. This gender imbalance underscores the need for future research to include a more balanced representation of genders. Furthermore, the linear regression models used in this study included variables derived from the ANOVA analyses. This choice was necessitated by the lack of prior literature directly addressing these clinical groups and their unique psychological profiles. While this exploratory approach allowed us to identify significant predictors, it may have limited the theoretical basis for interpreting the regression models. Future studies should aim to build on these findings, using larger samples and additional theoretical frameworks to refine and validate the models. While these limitations highlight areas for improvement, they also underscore the exploratory nature of this study and the need for further research. Despite these constraints, the findings provide valuable insights into the distinct psychological and clinical profiles of adolescents with headaches, restrictive eating disorders, and NSSI, offering a foundation for future studies and interventions.

## Figures and Tables

**Figure 1 pediatrrep-17-00021-f001:**
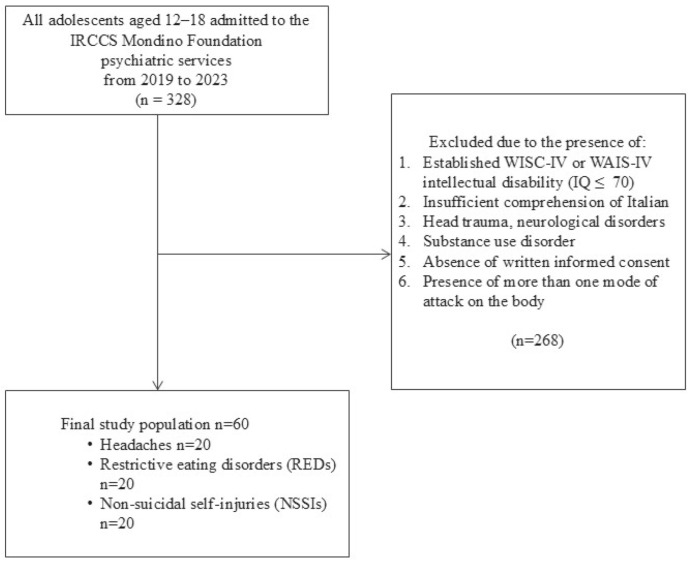
Study population flowchart. [authors’ own processing].

**Figure 2 pediatrrep-17-00021-f002:**
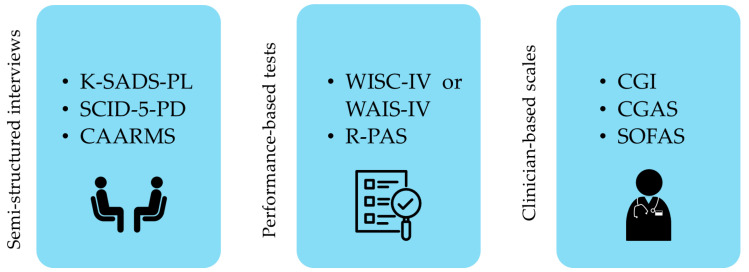
Research process. [authors’ own processing].

**Table 1 pediatrrep-17-00021-t001:** Comparisons’ significances regarding psychodiagnostics instruments in the three groups and post hoc analyses. [authors’ own processing].

Variables					a–b	a–c	b–c	Contrast
		F	*p*	ω^2^	*p*	*p*	*p*	
K-SADS-PL	Depression	9.940	<0.001 ***	0.230	**0.023 ***	**<0.001 *****	0.761	a < b; a < c
	Psychosis	3.279	0.045 *	0.071	0.261	**0.050 ***	1.000	a < c
	Social anxiety	5.987	0.004 **	0.143	0.329	**0.004 ****	0.329	a < c
	OCD	3.279	0.045 *	0.071	1.000	0.261	**0.050 ***	b > c
	AN	25.333	<0.001 ***	0.448	**<0.001 *****	0.206	**<0.001 *****	a < b; b > c
IQ	WMI	8.117	0.017 *	0.040	0.582	**0.013 ***	0.392	a < c
SCID-5 PD ^a^	Borderline	9.274	<0.001 **	0.252	1.000	**0.001 *****	1.000	a < c
	Negative Symptoms	8.398	<0.001 ***	0.243	**0.031 ***	**0.001 *****	0.904	a < b; a < c
	CGI-S	16.300	<0.001 ***	0.338	**0.003 ****	**<0.001 *****	0.491	a < b; a < c
	CGAS	5.686	0.006 **	0.135	0.565	**0.004 ****	0.144	a > c
	SOFAS	5.005	0.010 **	0.118	1.000	**0.015 ****	**0.045 ***	a > c; b > c

Significance: * = *p* < 0.05; ** = *p* < 0.01; *** = *p* < 0.001; ω^2^: very small (<0.01), small (between ≤0.01 and <0.06), medium (between ≤0.06 and <0.14), and large effect (≥0.14). Note: ^a^ only administered to patients 14 y.o. and older. Groups: a = headaches; b = restrictive eating disorders; c = NSSI. Abbreviations: AN: anorexia nervosa; CGAS: Children’s Global Assessment Scale; CGI-S: Clinical Global Impression Severity; OCD: obsessive–compulsive disorder; SOFAS: Social and Occupational Functioning Assessment Scale; WMI: working memory index.

**Table 2 pediatrrep-17-00021-t002:** Comparisons between the R-PAS variables in the three groups and post hoc analyses. [authors’ own processing].

				a–b	a–c	b–c	Contrast
	F	*p*	ω^2^	*p*	*p*	*p*	
PHR/GPHR ^1^	5.098	**0.009 ****	0.132	**0.014 ***	**0.044 ***	1.000	a < b; a < c
W% ^2^	4.855	**0.011 ***	0.114	**0.013 ***	0.082	1.000	a > b
Dd% ^3^	15.64	**<0.001 *****	0.328	**<0.001 *****	**0.002 ****	0.168	a < b; a < c
Cblend ^4^	8.318	**0.016 ***	0.063	0.688	0.284	**0.012 ***	b < c
C′ ^5^	6.633	**0.036 ***	0.088	1.000	0.267	**0.035 ***	b > c
NPH/SumH ^6^	4.475	**0.016 ***	0.114	0.093	1.000	**0.020 ***	b > c
PER ^7^	8.203	**0.017 ***	0.095	1.000	**0.047 ***	**0.033 ***	a < c; b < c

Significance: * = *p* < 0.05; ** = *p*< 0.01; *** = *p*< 0.001; ω^2^: very small (< 0.01), small (between ≤0.01 and <0.06), medium (between ≤0.06 and <0.14), and large effect (≥0.14). Groups: a = headaches; b = restrictive eating disorders; c = NSSI. Note: ^1^ Proportion of poor or good human representation; ^2^ whole%; ^3^ unusual detail%; ^4^ sum of the blend of achromatic/light dark and color determinants; ^5^ achromatic color; ^6^ proportion of not pure H responses; ^7^ personal responses.

**Table 3 pediatrrep-17-00021-t003:** Summary of the linear regression models. [authors’ own processing].

Model	R	R^2^	Adjusted R^2^	Std. Error of the Estimate					
					ΔR^2^	F Change	df1	df2	*p*
1	0.561 ^a^	0.315	0.296	0.638	0.315	16.109	1	35	**<0.001 ****
2	0.800 ^b^	0.639	0.618	0.470	0.324	30.574	1	34	**<0.001 ****
3	0.829 ^c^	0.688	0.659	0.444	0.048	5.101	1	33	**0.031 ***

Significance: * = *p* < 0.05; ** = *p* < 0.01; Predictors: ^a^. Negative symptoms. ^b^. Negative symptoms, SCID-5 PD borderline personality traits. ^c^. Negative symptoms. SCID-5 PD borderline personality traits, K-SADS-PL psychosis symptoms. Outcome variable: group.

**Table 4 pediatrrep-17-00021-t004:** Results of stepwise linear regression for predicting group membership.

Model	Variable	B	Standard Error	β	t	*p*
1	Constant	0.733	0.165	-	4.451	**<0.001 ****
	Negative symptoms	0.858	0.214	0.561	4.014	**<0.001 ****
2	Constant	0.273	0.147	-	1.854	0.072
	Negative symptoms	0.965	0.158	0.632	6.087	**<0.001 ****
	SCID-5 PD borderline	0.863	0.156	0.574	5.529	**<0.001 ****
3	Constant	0.198	0.143	-	1.382	0.176
	Negative symptoms	0.944	0.150	0.618	6.293	**<0.001 ****
	SCID-5 PD borderline	0.783	0.152	0.520	5.155	**<0.001 ****
	K-SADS-PL psychosis symptoms	0.355	0.157	0.226	2.258	**0.031 ***

Dependent variable: group. Predictors: negative symptoms, SCID-5 PD borderline personality traits, K-SADS-PL psychosis symptoms. Significance: * = *p* < 0.05; ** = *p* < 0.01.

## Data Availability

The data underlying this article are available upon reasonable request in Zenodo [24] (10.5281/zenodo.13838103).

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
