# Peer review of "Look at My Body: It Tells of Suffering—Understanding Psychiatric Pathology in Patients Who Suffer from Headaches, Restrictive Eating Disorders, or Non-Suicidal Self-Injuries (NSSIs)"

_pediatrrep, 2025, doi:10.3390/pediatric17010021_

Round 1
Reviewer 1 Report
Comments and Suggestions for Authors
The paper titled "Look at my body: it tells of suffering. Comprehending psychiatric pathology in patients who suffer from headaches, restrictive eating disorders (REDs), or nonsuicidal self-injuries (NSSIs)" has several strong elements and areas for improvement to consider for publication.
Below is my detailed review and revision recommendations:
1. Introduction
- Condense the background information and focus on the unique aspects of the study, such as the use of R-PAS in this context.
- Avoid repeating well-known facts about adolescence and psychopathology.
- Provide more detail on the rationale for using R-PAS compared to other tools.
4. Discussion
- Emphasize the clinical significance of the findings over the methodological details.
- Discuss limitations more critically, including potential biases introduced by the small, predominantly female sample.
- Propose more concrete steps for how the findings could inform clinical practice or future research.
Comments on the Quality of English Language
Fine
Author Response
Comment: The paper titled "Look at my body: it tells of suffering. Comprehending psychiatric pathology in patients who suffer from headaches, restrictive eating disorders (REDs), or nonsuicidal self-injuries (NSSIs)" has several strong elements and areas for improvement to consider for publication.
Below is my detailed review and revision recommendations:
Reply: We appreciated the time the reviewer took to read our work and give valuable comments.
Comment: 1. Introduction
Condense the background information and focus on the unique aspects of the study, such as the use of R-PAS in this context.
Avoid repeating well-known facts about adolescence and psychopathology.
Provide more detail on the rationale for using R-PAS compared to other tools.
Reply: We thank the reviewer for this suggestion. We updated the Introduction providing information about the R-PAS and the choice of including it in the study.
Comment: 4. Discussion
Emphasize the clinical significance of the findings over the methodological details.
Discuss limitations more critically, including potential biases introduced by the small, predominantly female sample.
Propose more concrete steps for how the findings could inform clinical practice or future research.
Reply: We appreciate the reviewer’s comments regarding Discussion improvement. We further discussed limitations, biases and future research and deepened the clinical significance of the study.
Reviewer 2 Report
Comments and Suggestions for Authors
Pratile et al present a study on the exploration of adolescents' functioning and personality traits with these conditions through a multimethod assessment.
They consider some of the medically unexplained symptoms that they do not characterize or discuss as such. They are strongly suggested to discuss accordingly.
Author Response
REVIEWER 2
Comment: Pratile et al present a study on the exploration of adolescents' functioning and personality traits with these conditions through a multimethod assessment.
They consider some of the medically unexplained symptoms that they do not characterize or discuss as such. They are strongly suggested to discuss accordingly.
Reply: We thank the reviewer for taking the time to read our work and giving valuable suggestions. We updated the manuscript accordingly.
Reviewer 3 Report
Comments and Suggestions for Authors
Title: do you really mean ‘comprehending’ or rather ‘understanding’?
Abstract: please report participants’ mean age and gender-ratio.
“… psychological changes that shape identity.”, should be: associated with shaping identity in a bi-directional fashion (Siegler et al., 2019)
“These self-directed physical harm conditions”; do the authors claim that headache is self-directed?
‘biological interactions’, though, your study is cross-sectional, yes?
Do you really mean ‘confront’ or rather ‘compare’?
assessment. (full stop, not semicolon).
Consider that you assessed adolescent individuals and not patients, or were all individuals hospitalized at the timepoint of assessment?
‘headache patients’; consider that nobody should be labeled based on a health issue and ‘headache patients’ do not exist, but individuals with (frequent) headache.
‘during the Rorschach’, do you mean ‘applying the Rorschach test’?
“The study reveals distinct psychiatric profiles among adolescents with self-harm 30 tendencies and highlights…”, please clearly show evidence that headache is a strategy of self-directed and deliberate self-harm.
It remains unclear why this sample was diagnosed, compared with and highlighted.
Introduction: Ref 1 is completely inappropriate; it is written in a language not for the standard reader of an international and peer-reviewed journal; a manual on psychotherapy is not an appropriate source of developmental psychology; a book is never thoroughly peer-reviewed; Aliprandi, Pelanda and Senise support a psychoanalytic approach, which by definition is not a scientifically proven and evidence-based psychotherapeutic intervention. More specifically, while the psychoanalytic approach systematically avoids its scientific investigation to explain, if, and if so, their approach might work, they also make it systematically impossible that their intervention is harmful (Grawe K. Neuropsychotherapy: How the Neurosciences Inform Effective Psychotherapy (Counseling and Psychotherapy). Routledge, New York NY, USA, 2006.
Data availability statement; the link is not valid.
Introduction:
“…This is the case of headaches, Restrictive Eating Disorders (REDs), and Non-Suicidal Self-Injury (NSSI) [2– 4], which we decided to focus on.” A closer look to the references revealed that none of them focused on headache or self-injury. Is it possible that the authors of the present study were overconfident as regards the use of the references? Relatedly, a quick glance to the DSM-5 (American Psychiatric Association, 2013) and DSM-5 TR (American Psychiatric Association, 2022) showed that restrictive eating disorder is not a nosological category of its own; please clarify and add the references.
“…. distress is increasingly expressed through REDs…”; what do you mean with ‘increasingly’?
“…. Similarly, self-harming behaviors are on the rise among adolescents.” please add references.
“…Furthermore, the influence of social media on NSSI is evident”; please add references.
“In addition to EDs and NSSI, psychological distress in adolescents is sometimes expressed through somatization, with headaches being one of the most common manifestations.” please add references.
“.. the Rorschach inkblot test is one of the most promising performance-based tests for comprehending patients’ functioning and personality characteristics”, please add references.
Please formulate hypotheses and describe in more details, if and to what extent the present study might add to the current literature in an important fashion.
Method; as mentioned above, cautiously use the label ‘patient’.
Inclusion and exclusion criteria; how come nobody had ADHD or anxiety disorders, autism spectrum disorder, PTDS or symptoms of schizophrenia spectrum disorder? Otherwise, the assessment was thoroughly performed.
Statistics: please report effect sizes, which are robust against sample sizes.
Results; for such small sample sizes, the number of tests is far too large, even if correcting p-values for multiple testing. Rather, consider to run a logistic regression model to identify significant predictors.
Discussion: please clearly answer the hypotheses, and substantively enlarge the Limitation section.
Conclusions; this is often a question of taste and judgement, though, trim this section to the maximum.
References
American Psychiatric Association, 2013. Diagnostic and Statistical Manual of Mental Disorders 5th edition: DSM 5. American Psychiatric Association, Arlington VA.
American Psychiatric Association, 2022. Diagnostic and statistical manual of mental disorders (DSM-5 TR), 5th edition, text revision ed. American Psychiatric Association, Arlington WA, USA.
Siegler, R., Saffran, J.R., Gershoff, E.T., Eisenberg, N., 2019. How children develop, Sixth edition ed. Worth Publishers; Macmillan Learning, New York, NY, USA.
Comments on the Quality of English Language
Please be careful not to use Italian expressions as English expressions.
Author Response
Comment: Title: do you really mean ‘comprehending’ or rather ‘understanding’?
Reply: After an extensive English revision, we agree with the reviewer’s comment.
Comment: Abstract: please report participants’ mean age and gender-ratio.
Reply: We thank the reviewer for raising this point. We reported the required data in the Abstract.
Comment: “… psychological changes that shape identity.”, should be: associated with shaping identity in a bi-directional fashion (Siegler et al., 2019)
Reply: We extensively updated the abstract to make it more readable.
Comment: “These self-directed physical harm conditions”; do the authors claim that headache is self-directed?
Reply: Thank you for your thoughtful comment. We truly appreciate the opportunity to clarify this point. We aimed to highlight that headaches appear as a way for adolescents to express psychological distress through somatic symptoms. Recent studies on psychosomatic disorders in adolescents support this view, highlighting the complex relationship between psychological stress and somatic symptoms like headaches.
Comment: ‘biological interactions’, though, your study is cross-sectional, yes?
Reply: We thank the reviewer for having reported this error. We corrected it to avoid misunderstandings.
Comment: Do you really mean ‘confront’ or rather ‘compare’?
Reply: We revised the manuscript to make it more precise and readable.
Comment: assessment. (full stop, not semicolon).
Reply: We updated this.
Comment: Consider that you assessed adolescent individuals and not patients, or were all individuals hospitalized at the timepoint of assessment?
Reply: We performed the research in a third-level center that admits only adolescents with severe symptoms. Accordingly, they could all be considered patients because they were admitted to our Institute inpatients or day hospital regimens. We reported this in the study population flowchart and the study population paragraph.
Comment: ‘headache patients’; consider that nobody should be labeled based on a health issue and ‘headache patients’ do not exist, but individuals with (frequent) headache.
Reply: We thank the reviewer for raising this. We updated the manuscript to make it more precise.
Comment: ‘during the Rorschach’, do you mean ‘applying the Rorschach test’?
Reply: We appreciated the reviewer’s comment and modified the text to clarify it.
Comment: “The study reveals distinct psychiatric profiles among adolescents with self-harm 30 tendencies and highlights…”, please clearly show evidence that headache is a strategy of self-directed and deliberate self-harm. It remains unclear why this sample was diagnosed, compared with and highlighted.
Reply: We followed the reviewer’s suggestion and better explained the relationship between headaches and physical symptoms. Thank you for your valuable feedback. We followed your suggestion and clarified the relationship between headaches and physical symptoms. Headaches in adolescence can be the expression of psychological distress and emotional difficulties throughout the body. These issues are often present in highly sensitive and emotional individuals who may struggle to express their emotions and needs. In addition, headaches in adolescents are frequently associated with psychological conditions such as anxiety and depression. While a headache is not an intentional self-injurious act, like cutting, it can still represent a form of expressing psychological distress through the body. This underscores the need for an integrated approach that considers the physical and emotional aspects of treating adolescents with headaches.
Comment: Introduction: Ref 1 is completely inappropriate; it is written in a language not for the standard reader of an international and peer-reviewed journal; a manual on psychotherapy is not an appropriate source of developmental psychology; a book is never thoroughly peer-reviewed; Aliprandi, Pelanda and Senise support a psychoanalytic approach, which by definition is not a scientifically proven and evidence-based psychotherapeutic intervention. More specifically, while the psychoanalytic approach systematically avoids its scientific investigation to explain, if, and if so, their approach might work, they also make it systematically impossible that their intervention is harmful (Grawe K. Neuropsychotherapy: How the Neurosciences Inform Effective Psychotherapy (Counseling and Psychotherapy). Routledge, New York NY, USA, 2006.
Reply: We understood the reviewer’s comment and revised the manuscript to make it more readable, enriching it with more appropriate references.
Comment: Data availability statement; the link is not valid.
Reply: As we know, the repository website will make the dataset available after the revision process. Once the publication process is completed, the DOI assigned to the manuscript will be linked to the DOI assigned to the dataset.
Comment: Introduction:
“…This is the case of headaches, Restrictive Eating Disorders (REDs), and Non-Suicidal Self-Injury (NSSI) [2– 4], which we decided to focus on.” A closer look to the references revealed that none of them focused on headache or self-injury. Is it possible that the authors of the present study were overconfident as regards the use of the references? Relatedly, a quick glance to the DSM-5 (American Psychiatric Association, 2013) and DSM-5 TR (American Psychiatric Association, 2022) showed that restrictive eating disorder is not a nosological category of its own; please clarify and add the references.
Reply: We thank the reviewer for raising this point. Although restrictive eating disorder is not a DSM nosological category, this term is widely used in the field, and our research group has been using it in publications since 2021. The term well describes the patients referred to the Eating Disorders Center of our institute, which takes care of patients who present with very severe restrictive eating symptoms that cannot always be traced to a nosographic label. Moreover, as the reviewer suggested, we revised the references to cite only the appropriate papers.
Comment: “…. distress is increasingly expressed through REDs…”; what do you mean with ‘increasingly’?
Reply: We made an extensive revision to make the paper more readable.
Comment: “…. Similarly, self-harming behaviors are on the rise among adolescents.” please add references.
Reply: We followed the reviewers’ suggestions and extensively updated the Introduction.
Comment: “…Furthermore, the influence of social media on NSSI is evident”; please add references.
Reply: We followed the reviewers’ suggestions and extensively modified the Introduction.
Comment: “In addition to EDs and NSSI, psychological distress in adolescents is sometimes expressed through somatization, with headaches being one of the most common manifestations.” please add references.
Reply: We followed the reviewers’ suggestions and rewrote the Introduction.
Comment: “.. the Rorschach inkblot test is one of the most promising performance-based tests for comprehending patients’ functioning and personality characteristics”, please add references.
Reply: We added the appropriate references, also explaining the potential of the Rorschach test using the R-PAS.
Comment: Please formulate hypotheses and describe in more details, if and to what extent the present study might add to the current literature in an important fashion.
Reply: We appreciated the reviewer’s comment. We addressed it and improved the manuscript.
Comment: Method; as mentioned above, cautiously use the label ‘patient’.
Reply: We explained above the choice of "patient" term.
Comment: Inclusion and exclusion criteria; how come nobody had ADHD or anxiety disorders, autism spectrum disorder, PTDS or symptoms of schizophrenia spectrum disorder? Otherwise, the assessment was thoroughly performed.
Reply: We thank the reviewer for this comment. We excluded participants presenting comorbid diagnoses from the other groups to reduce biases, but not patients with other comorbidities. The clinical interviews and the assessment did not reveal symptoms of such severity that a diagnosis of ADHD, autism spectrum disorder, PTSD, or schizophrenia could be made. In Table 1, we reported the results of the semi-structured interview K-SADS showing the presence of symptomatology for each group of participants regarding the DSM-5 diagnoses.
Comment: Statistics: please report effect sizes, which are robust against sample sizes.
Reply: We reported the effect sizes, as requested.
Comment: Results; for such small sample sizes, the number of tests is far too large, even if correcting p-values for multiple testing. Rather, consider to run a logistic regression model to identify significant predictors.
Reply: We thank the reviewer for raising this issue. Due to the small sample size, which was further reduced because of the division into groups, we thought logistic regression analysis might not provide reliable results. However, to address the reviewer’s suggestion, we converted K-SADS and SCID-5 PD categorical variables into dummy variables (0: absence of symptoms, 1: presence of symptoms) and ran a linear regression. Because of the absence of previous research supporting our hypotheses, we used the ANOVA statistically significant variables to run the model. In the proper section, we added this limitation connected to the numerosity and the sample’s specificity. We explained this choice in the Statistical analysis paragraph.
Comment: Discussion: please clearly answer the hypotheses, and substantively enlarge the Limitation section.
Reply: We thank the reviewer for the suggestion. We updated the Discussion section accordingly.
Comment: Conclusions; this is often a question of taste and judgement, though, trim this section to the maximum.
Reply: We addressed the reviewer’s request and removed the Conclusions.
Comment: References
American Psychiatric Association, 2013. Diagnostic and Statistical Manual of Mental Disorders 5th edition: DSM 5. American Psychiatric Association, Arlington VA.
American Psychiatric Association, 2022. Diagnostic and statistical manual of mental disorders (DSM-5 TR), 5th edition, text revision ed. American Psychiatric Association, Arlington WA, USA.
Siegler, R., Saffran, J.R., Gershoff, E.T., Eisenberg, N., 2019. How children develop, Sixth edition ed. Worth Publishers; Macmillan Learning, New York, NY, USA.
Reply: We checked all the references and made the proper updates.
Reviewer 4 Report
Comments and Suggestions for Authors
This is an interesting survey. Psychopathological analysis of patients with headache, restrictive eating disorder (RED) or non-suicidal self-injury (NSSI). I will raise the following points and questions about the content of the manuscript to provide the author with reference and improvements.
1. Summary
The author needs to simplify the abstract narrative.
2. Introduction
Themes and objects that reinforce the focus of this manuscript.
Citations need to be updated.
For example:
https://doi.org/10.1080/10640266.2023.2196492
https://www.proquest.com/openview/0b7c348b303acf4eae0d3b9b92fe4cb3/1?pq-origsite=gscholar&cbl=18750&diss=y
3.Materials and methods
Regarding the study population, although the author describes your research process, this is not conducive to reading. It is recommended that the author draw a flow chart of your experiment based on the narrative content.
4.Results
Although the authors want to make efforts to express the data results. Unfortunately, the table is too long, not conducive to reading, and fails to arouse readers' interest.
Suggest adjustments or modifications.
5. Discussion
It's great that the author actively and diligently explains the answers derived from the data. However, for the knowledgeable reader, the results of these data can be understood from tables or figures. This seems to lose the meaning of data analysis and discussion.
It is recommended that the author provide personal insights based on the analysis results to increase the academic contribution that the author hopes to reflect through the results of this manuscript.
I look forward to further reading after the author provides a revised manuscript,
Author Response
Comment: This is an interesting survey. Psychopathological analysis of patients with headache, restrictive eating disorder (RED) or non-suicidal self-injury (NSSI). I will raise the following points and questions about the content of the manuscript to provide the author with reference and improvements.
- Summary
The author needs to simplify the abstract narrative.
Reply: We thank the reviewer for raising this point. We modified the Abstract as requested.
Comment: 2. Introduction
Themes and objects that reinforce the focus of this manuscript.
Citations need to be updated.
For example:
https://doi.org/10.1080/10640266.2023.2196492
https://www.proquest.com/openview/0b7c348b303acf4eae0d3b9b92fe4cb3/1?pq-origsite=gscholar&cbl=18750&diss=y
Reply: We thank the reviewer for those valuable suggestions. We updated the Introduction to make it more incisive and readable.
Comment: 3.Materials and methods
Regarding the study population, although the author describes your research process, this is not conducive to reading. It is recommended that the author draw a flow chart of your experiment based on the narrative content.
Reply: We thank the reviewer and addressed the comment by including a figure named Figure 2 representing the research process.
Comment: 4.Results
Although the authors want to make efforts to express the data results. Unfortunately, the table is too long, not conducive to reading, and fails to arouse readers' interest.
Suggest adjustments or modifications.
Reply: We thank the reviewer for this suggestion. We extensively updated the tables, highlighting only relevant results and moving all others into supplementary tables for completeness.
Comment: 5. Discussion
It's great that the author actively and diligently explains the answers derived from the data. However, for the knowledgeable reader, the results of these data can be understood from tables or figures. This seems to lose the meaning of data analysis and discussion.
It is recommended that the author provide personal insights based on the analysis results to increase the academic contribution that the author hopes to reflect through the results of this manuscript.
I look forward to further reading after the author provides a revised manuscript
Reply: We thank the reviewer for raising this point and helping us improve the manuscript. We extensively review the Discussion.
Round 2
Reviewer 3 Report
Comments and Suggestions for Authors
While the Reviewer appreciates the authors’ efforts to improve the quality of the manuscript, several flaws related to the study per se and to the manuscript could not be fixed.
Even if the same research group used the expression of Restrictive Eating Disorder (RED) in a previous publication, this does not implicitly officialize the expression as a nosological entity.
Such idiosyncratic expressions lead to scientific misunderstandings.
The authors still need to explain, why they have chosen a psychoanalytic approach, which has no scientific basis.
The authors have noteworthy improved the Introduction section, though hypotheses are still missing, and the authors should explain in more details, if and to what extent the present study adds to the current knowledge in an important fashion. Please consider that formulating study aims does not equal to formulating hypotheses.
To report effect sizes with Omega squared was particularly smart, though, the reader needs guidance as regards the interpretation and the cut-off values. Further, the reference of Kroes and Finley is incomplete.
The wording is still politically and clinically incorrect; there are no NSSI patients, but only patients with NSSI.
Even though the authors introduced nicely the effect sizes, those measures were not further considered when discussing their results.
Author Response
Comment: the Reviewer appreciates the authors’ efforts to improve the quality of the manuscript, several flaws related to the study per se and to the manuscript could not be fixed.
Response: We thank the reviewer for the comments. They have made the article more valuable despite some limitations partly due to the exploratory nature of the study, which we have considered, and need to be addressed in further research.
Comment: Even if the same research group used the expression of Restrictive Eating Disorder (RED) in a previous publication, this does not implicitly officialize the expression as a nosological entity. Such idiosyncratic expressions lead to scientific misunderstandings.
Response: The reviewer raised a good point. We have updated the part where we better explain the characteristics of the patients belonging to the group called “RED” in the study. We have specified that this definition does not refer to a DSM nosological category, but is a term adopted by the authors and in other works in the literature to include patients who presented an ED with restrictive characteristics. We modified as follows: “The second group comprised patients presenting a diagnosis of any eating disorder with restrictive characteristics. We included restrictive subtypes of anorexia nervosa (AN), atypical anorexia nervosa (A-AN), avoidant/restrictive food intake disorder (ARFID), or other specified eating disorders with restrictive characteristics. All the diagnoses were made according to the Diagnostic and Statistical Manual of Mental Disorders (DSM-5) criteria [citation].” (lines 124-129)
Comment: The authors still need to explain, why they have chosen a psychoanalytic approach, which has no scientific basis.
Response: We addressed the reviewer’s suggestions and made the text more accessible for the standard reader of international and peer-reviewed journals, updating the references and deleting the psychoanalytic references related to development in adolescence. The research design and the tools used are not influenced by the psychoanalytic orientation and are validated for research.
Comment: The authors have noteworthy improved the Introduction section, though hypotheses are still missing, and the authors should explain in more details, if and to what extent the present study adds to the current knowledge in an important fashion. Please consider that formulating study aims does not equal to formulating hypotheses.
Response: We appreciated the reviewer’s suggestion. We added our hypotheses at the end of the Introduction. “Considering previous research and theoretical considerations, we hypothesized that adolescents engaging in NSSI would exhibit the lowest levels of global functioning, given the strong association between self-injurious behaviors and broader psychosocial impairments. Moreover, we would expect that the group affected by RED will display higher IQ scores than the other groups, potentially reflecting the perfectionism and achievement-oriented traits often associated with this population. However, we also expected this group to show significant impairments in self and other representation, as interpersonal difficulties are commonly observed in adolescents with eating disorders. Finally, we proposed that adolescents suffering from headaches would demonstrate greater rigidity in emotional expression and regulation strategies. This expectation was based on the idea that somatic symptoms, such as headaches, may act as a manifestation of unexpressed or poorly regulated emotional states.” (lines 90-101)
Comment: To report effect sizes with Omega squared was particularly smart, though, the reader needs guidance as regards the interpretation and the cut-off values. Further, the reference of Kroes and Finley is incomplete.
Response: We added in the manuscript the cut-offs of the omega squared effect size. “Thresholds indicate very small (< 0.01), small (between ≤ 0.01 and <0.06), medium (between ≤ 0.06 and < 0.14) and large effect (≥ 0.14).” Moreover, we checked the correctness of the reference, and we confirm that the proper citation suggested by the authors is “Kroes, A. D. A., & Finley, J. R. (2023, July 20). Demystifying Omega Squared: Practical Guidance for Effect Size in Common Analysis of Variance Designs. Psychological Methods. Advance online publication. https://dx.doi.org/10.1037/met0000581”. We updated the reference in the manuscript. (RIGHE)
Comment: The wording is still politically and clinically incorrect; there are no NSSI patients, but only patients with NSSI.
Response: We thank the reviewer for this valuable comment. We updated the whole manuscript to avoid political and clinical incorrect wording.
Comment: Even though the authors introduced nicely the effect sizes, those measures were not further considered when discussing their results.
Response: We addressed the reviewer’s comment, adding the relevance of effect sizes in the Discussion.
Reviewer 4 Report
Comments and Suggestions for Authors
We thank the authors for submitting the revised manuscript. I think this edited manuscript has clearly been improved based on the comments and is of sufficient quality to move forward. It is suggested that the editor-in-chief may consider adopting this revised manuscript for the next stage of review and evaluation.
Author Response
Comment: We thank the authors for submitting the revised manuscript. I think this edited manuscript has clearly been improved based on the comments and is of sufficient quality to move forward. It is suggested that the editor-in-chief may consider adopting this revised manuscript for the next stage of review and evaluation.
Response: We thank the reviewer for the appreciation and the valuable comments and recommendations. Following the reviewers’ suggestions, we modified the manuscript to make it more compelling and complete.
Round 3
Reviewer 3 Report
Comments and Suggestions for Authors
The authors are getting closer to the final stretch! Congrats on you! Please carefully go through the text and systematically erase the acronym RED "..at the group affected by RED will display higher IQ .."
Author Response
Comment: The authors are getting closer to the final stretch! Congrats on you! Please carefully go through the text and systematically erase the acronym RED "..at the group affected by RED will display higher IQ .."
Response: We addressed the reviewer’s comment carefully, going through the text to systematically remove the acronym "RED" as suggested.